# Continuous Oral Administration of Sonicated *P. gingivalis* Delays Rat Skeletal Muscle Healing Post-Treadmill Training

**DOI:** 10.3390/ijerph192013046

**Published:** 2022-10-11

**Authors:** Kairi Hayashi, Yasuo Takeuchi, Shintaro Shimizu, Gen Tanabe, Hiroshi Churei, Hiroaki Kobayashi, Toshiaki Ueno

**Affiliations:** 1Department of Masticatory Function and Health Science, Graduate School of Medical and Dental Science, Tokyo Medical and Dental University, Tokyo 113-8510, Japan; 2Division of Sports Dentistry of Sports Science Organization, Tokyo Medical and Dental University, Tokyo 113-8510, Japan; 3Department of Periodontology, Graduate School of Medical and Dental Science, Tokyo Medical and Dental University, Tokyo 113-8510, Japan; 4Department of Oral Microbiology, Asahi University School of Dentistry, Gifu 501-0296, Japan; 5Department of Sports Dentistry, Meikai University School of Dentistry, Saitama 350-0283, Japan

**Keywords:** *P. gingivalis*, skeletal muscle, training, satellite cell

## Abstract

Background: A delay in muscle repair interferes with the effect of training or exercise; therefore, it is important to identify the factors that delay muscle repair. *P. gingivalis*, one of the most common periodontal disease pathogens, has the potential to inhibit muscle repair after training, as inferred from a previous study. To assess the expression of satellite cells in this in vivo study, we evaluated the relationship between *P. gingivalis* and muscle regeneration after training. Methods: A total of 20 male Wistar rats (eight weeks in age) were randomly divided into two groups: one orally administered sonicated *P. gingivalis* four times per week for six weeks (PG group) and one given no treatment (NT group). After four weeks of training using a treadmill, the gastrocnemius was evaluated using histology of the cross-sectional area (CSA) of myotubes and immunohistochemistry of the expression of skeletal muscle satellite cells. In addition, an endurance test was performed a day before euthanization. Results: The CSA and expression of Pax7+/MyoD− and Pax7+/MyoD+ cells were not significantly different between the groups. However, the expression of Pax7−/MyoD+ cells and running time until exhaustion were significantly lower in the PG group. Conclusions: Infection with *P. gingivalis* likely interferes with muscle repair after training.

## 1. Introduction

Training or exercise is performed for various reasons, including the maintenance of good health and the improvement of athletic ability. However, hard training often causes skeletal muscle injury. A certain period of time is necessary to allow for skeletal muscle repair, but if this is delayed, muscle hypofunction can occur, decreasing the health and level of performance. Hence, continuous repair of damaged muscles after hard training is required to maintain high performance. 

Satellite cells are located outside the sarcolemma and under the basal lamina of muscle fibers [1]. After a muscle injury, satellite cells are activated and divide, which leads to the formation of new myofibers or the repair of existing ones [2]. Satellite cells express the transcription factor paired box protein 7 (Pax7) in the quiescent state, and when they are activated, the myogenic differentiation protein 1 (MyoD) is induced and co-expressed with Pax7. As differentiation progresses, Pax7 is downregulated, and MyoD stimulates further differentiation into the myoblast lineage [3,4]. If this process is delayed, smooth muscle repair also becomes delayed. Therefore, it is necessary to detect the underlying mechanisms to use as potential targets to treat this.

We hypothesized that oral bacteria hinder muscle repair after hard training. Oral bacteria related to periodontal disease have been reported to affect not only oral health but also systemic diseases, such as diabetes, cardiovascular disease, and low birth weight [5,6,7,8,9,10]. One of the most important pathogens of periodontal disease, *Porphyromonas gingivalis* (*P. gingivalis*), invades blood vessels from the endothelial cells of the periodontal pocket; bacterial toxins and their components, such as lipopolysaccharide (LPS, an endotoxin), travel through the bloodstream [11,12]. Recent studies have suggested that *P. gingivalis* is a “keystone” of oral microbial dysbiosis, in which *P. gingivalis* acts as a critical agent by disrupting host immune homeostasis [13], possibly leading to patient periodontitis. *P. gingivalis* is also present in adolescents and young adults without periodontal disease and does not lead to periodontitis independently [14]. Therefore, when discussing the effect of periodontitis, *P. gingivalis* should be placed in a broader context, such as the Socransky complex and periodontal microbiota with the passage from symbiosis to dysbiosis. However, some conditions, such as bad oral health, can accelerate gingivitis or bone loss, resulting in an increase of oral *P. gingivalis* [15] and causing many bacterial toxins and components to spread to the whole body through the bloodstream. Therefore, our research focus was on the effect of *P. gingivalis*, specifically the spread of *P. gingivalis* components, for example, LPS, leading to the uptake of inflammatory cytokines, such as interleukin-6, tumor necrosis factor-alpha (TNF-α), and C-reactive proteins, in remote organs such as skeletal muscle [16,17]. For example, Zhang et al. reported that 41% of patients with periodontal disease (probing periodontal pocket depth ≥ 4 mm) had elevated serum C-reactive protein levels (≥5 mg/L) compared to 19% of non-periodontal disease patients [18]. Previous studies have reported that inflammatory cytokines negatively impact skeletal muscle regeneration [19,20]. These findings suggest that an increase in *P. gingivalis* will inhibit muscle repair after hard training. However, to our knowledge, this relationship has not yet been directly evaluated.

Hence, we used a rat model for in vivo oral administration of *P. gingivalis* and subjected the rats to high-intensity treadmill training. After training, skeletal muscle was examined, and the expression of Pax7 and MyoD was measured to evaluate the relationship between *P. gingivalis* and muscle regeneration.

This is the first study to analyze the relationship between *P. gingivalis* and skeletal muscle in vivo.

## 2. Materials and Methods

### 2.1. Animals

Twenty 8-week-old male Wistar rats were obtained from the Sankyo Labo Service Corporation, Inc. (Tokyo, Japan). Rats were randomly divided into two groups. Ten rats were orally administered sonicated *P. gingivalis*, and ten were assigned to the NT group and given no treatment. Both groups of rats were subjected to treadmill exercise and euthanized 24 h after the last running session.

All animal experiments were approved by the Institutional Animal Care and Use Committee of the Tokyo Medical and Dental University (A2020-136C4) and were housed under the same conditions and maintained on food and water ad libitum.

### 2.2. Induction of Experimental Periodontal Disease

*P. gingivalis* ATCC 33277 was anaerobically inoculated into a brain heart infusion broth medium (supplemented with 5 mg/L of hemin and 50 µg/L of vitamin K1) for two days at 35 °C. After sonication, the number of bacterial cells was counted using a bacterial counting chamber. The concentration was adjusted to 10^9^ cells/mL, and the solution was centrifuged (8000× *g*, 4 °C, 10 min). The pellet was resuspended in a physiological saline. The PG group rats were orally administered 200 µL of saline with sonicated *P. gingivalis* four times per week for six weeks.

### 2.3. Treadmill Training

After one week of oral administration, all rats were exercised for four days per week for four weeks on a motorized treadmill (Osaka Micro System Ltd., Osaka, Japan). Exercise intensities were 25 m/min for 30 min with a 5% slope in the first week. To avoid adaptation to training intensities, the running speed was increased by 2 m/min per week, and the training speed was 31 m/min in the last week (Figure 1).

### 2.4. Measurement of Body Weight and Muscle Wet Weight 

The body weights of the rats were measured before training and immediately before euthanization using a digital scale (GF-8K; A&D Co., Ltd., Tokyo, Japan). The wet weight of the gastrocnemius muscle was also measured immediately after the muscle was extracted.

### 2.5. Serum IgG Antibody Titers in Response to P. gingivalis

Immediately after the rats were euthanized using an overdose of carbon dioxide, blood samples were collected from the heart. Blood was allowed to clot at room temperature, and the serum was separated by centrifugation at 1500× *g* for 15 min. The serum was collected and stored at −80 °C until further analysis.

Serum IgG antibody titers were measured using an enzyme-linked immunosorbent assay (ELISA) [21,22]. Briefly, 96-well microplates (EIA plate; Costar, MA, USA) were coated with sonicated *P. gingivalis* ATCC 33277 at 10 µg/mL in a carbonate buffer, and a serially diluted reference positive control serum (2^5^–2^15^) and diluted serum (2^10^) were applied. Subsequently, phosphatase-conjugated goat anti-rat IgG (goat anti-rat IgG/H&L antibody, HRP conjugate; Protein Tech Japan, Tokyo, Japan) was added and developed with a phosphatase substrate. The optical density at 450 nm was measured using a microplate reader. The measured values were calculated as the ratio of the serum IgG antibody titer value of the control serum and averaged for each group.

### 2.6. Histological Analysis

The gastrocnemius muscles were embedded in paraffin after fixation with 4% paraformaldehyde for three days. Transverse sections of the gastrocnemius muscles were cut at 7 μm thickness using a rotary microtome (Microm HM325 Rotary Microtome; Thermo Fisher Scientific, Waltham, MA, USA) between the origin and insertion (Achilles tendon). The sections were stained with hematoxylin and eosin (H&E), and the slides were evaluated under a light microscope. Microphotographs were taken using a digital camera (Olympus AX70; Olympus, Tokyo, Japan) attached to a microscope (Olympus BX51; Olympus). We randomly chose nine high-power fields (HPFs) from each H&E-stained section and measured the cross-sectional area (CSA) of gastrocnemius muscle fibers using an image analysis software (ImageJ; National Institutes of Health, Bethesda, MD, USA). We traced the shapes of muscle fibers and calculated the CSA (µm^2^) using the known distance and detected pixel size. All measured CSA were averaged for each group.

### 2.7. Immunohistochemistry

Paraffin-embedded gastrocnemius muscles were cut at a 7-µm thickness using a rotary microtome (Microm HM325 Rotary Microtome; Thermo Fisher Scientific). The slides were activated for antigenicity using a citrate buffer solution and microwaved after removing the paraffin. After washing in PBS, the slides were immersed in a blocking solution (5% normal goat serum; Abcam, Cambridge, UK) for 30 min and then incubated overnight at 4 °C with the primary antibody (MyoD, mouse monoclonal antibody; Abcam; Pax7, mouse monoclonal antibody; Abcam), which was diluted 1:100 in PBS. The sections were washed in PBS three times each for 5 min and then incubated with secondary antibodies (goat anti-mouse IgG-Alexa Fluor 594, goat anti-rabbit IgG-Alexa Fluor 488; Abcam), which were diluted 1:400 in PBS for 1 h. The sections were then washed in PBS three times each for 5 min. Finally, the sections were mounted in a mounting solution with DAPI (Fluoroshield Mounting Medium with DAPI; Abcam). PAX7+/MyoD−, PAX7+/MyoD+, and PAX7−/MyoD+ cells were counted in nine HPFs from each slide. Each cell was calculated as a percentage of the total number of nuclei.

### 2.8. Endurance Test

The endurance was evaluated as running time until exhaustion, with the rat remaining on the stimulator of the treadmill for more than 5 s. The test was performed one day before euthanization and five days after the last training. The measured time was averaged for each group.

### 2.9. Statistical Analysis

The mean values of each were calculated, and statistical differences between the two groups were analyzed using the Mann−Whitney U test. All statistical analyses were performed using the Ekuseru−Toukei 2015 software (Social Survey Research Information Co., Ltd., Tokyo, Japan), and *p* < 0.05 was considered significant.

## 3. Results

### 3.1. Serum IgG Antibody Titers in Response to P. gingivalis

To examine the effect of *P. gingivalis* oral administration on the whole body, we analyzed serum IgG antibody titers in response to *P. gingivalis* from blood. 

In response to *P. gingivalis*, the serum IgG antibody titers in the PG and NT groups are shown in Figure 2. The PG group had significantly higher scores than the NT group (*p* = 0.0342). In the PG group, IgG antibody titers were approximately 30 times greater than those in the control serum. These data suggest that oral administration of *P. gingivalis* rose serum IgG titers in response to *P. gingivalis* in the blood of the whole body. 

IgG antibody titers of the PG group were significantly higher than those of the NT group.

### 3.2. Body Weight and Muscle Wet Weight

The body weight and muscle wet weight results are shown in Table 1. There was no significant difference between the PG and NT groups at either time point (*p* = 0.762 and 0.450, respectively). These data suggest that oral administration of sonicated *P. gingivalis* had no effect on dietary intake.

The wet weight of the isolated gastrocnemius muscle was measured immediately after euthanization, and there was no significant difference between the weights in either group (*p* = 0.820).

### 3.3. CSAs of Myofibers in H&E Staining

The mean (±SE) CSAs of the myofibers were 1159.0 ± 73.69 µm^2^ in the PG group and 1094.2 ± 72.71 µm^2^ in the NT group. There was no significant difference between the PG and NT groups (*p* = 0.450, Figure 3).

### 3.4. The Expression of Pax7 and MyoD in Immunohistochemistry

Immunohistochemical images of the expression of Pax7 and MyoD are shown in Figure 4. There was no significant difference in the number of Pax7- and MyoD-positive nuclei as total nuclei. The mean (±SE) rates of Pax7+/MyoD− nuclei in the PG and NT groups were 5.21 ± 0.77% and 5.63 ± 1.04%, respectively. Furthermore, Pax7+/MyoD+ nuclei in the PG and NT groups were 13.97 ± 1.18% and 14.64 ± 2.24%, respectively, and Pax7−/MyoD+ nuclei in the PG and NT groups were 4.62 ± 0.95% and 9.24 ± 0.86%, respectively. The rate of Pax7−/MyoD+ was significantly higher in the PG group (*p* = 0.00632). However, there was no significant difference in the rates of Pax7+/MyoD− and Pax7+/MyoD+ (*p* = 0.599 and 0.916, respectively). These data suggest that muscle regeneration was delayed in the PG group.

### 3.5. Time to Running Exhaustion

Mean (±SD) running time to exhaustion in the PG and NT groups was 2768.50 ± 550.79 s and 3406.75 ± 470.24 s, respectively, and significantly shorter in the PG group (*p* = 0.0274, Figure 5). These data suggest that rats in the PG group tired easily.

## 4. Discussion

This study aimed to evaluate the effects of *P. gingivalis,* one of the most common pathogens of periodontitis, on muscle degradation in vivo. We administered sonicated *P. gingivalis* (10^9^/mL) 200 µL/day for six weeks. An ELISA analysis showed that serum IgG antibody titers in response to *P. gingivalis* infection were higher in the PG group than in the NT group. It was considered that oral administration of *P. gingivalis* may cause it to invade the blood stream through the gingival crevice. Previous studies have reported that IgG antibody levels against *P. gingivalis* increase in patients with periodontitis [23,24]. The IgG antibody level against *P. gingivalis* in this study supported that oral administration of *P. gingivalis* affects the immune function. 

All rats were exercised using a treadmill, and both groups performed exercises of the same intensity. The exercise intensities used were between 25–31 m/min for 30 min with a 5% slope. The treadmill intensity used in the current study was chosen based on the evaluation of accumulated lactate levels in the blood in previous studies [25,26]. In these studies, an intensity of more than 25 m/min was considered high-intensity, and muscle damage was observed during this high-intensity training. The treadmill intensity in this study was higher than the intensity in previous studies; therefore, it was high enough to cause muscle damage. The exercise intensity in this study was assumed to be hard training, similar to training for athletes. 

The CSA of myofibers was not significantly different between the groups. However, the number of Pax7−/MyoD+ cells was significantly lower in the PG group than in the NT group. During muscle differentiation following muscle damage, Pax7 was downregulated, and MyoD was upregulated, indicating that satellite cell differentiation was delayed in the PG group. In other words, *P. gingivalis* infection may interfere with muscle repair after treadmill training in rats. Delayed muscle repair may affect muscle performance. Thus, in this study, running time until exhaustion was shorter in the PG group compared with the NT group. 

In patients with periodontitis, bacteremia usually occurs at the periodontium, accompanied by bacteria, LPS, and outer membrane protein. In previous studies, these led to an increase in inflammatory cytokines, such as interleukin-1, interleukin-6, and TNF-α, which, along with LPS, spread across the body through the blood [27,28]. After a muscle injury, the neutrophil influx is followed by the infiltration of macrophages with an M1 phenotype, which participates in satellite cell activation [29]. The increase in M1 macrophages is followed by the expansion of M2 macrophages, which are associated with satellite cell differentiation [30,31]. These macrophages modulate satellite cell activation and differentiation by modulating fibroadipogenic progenitor cells in the acute inflammatory response after muscle damage, which is associated with muscle regeneration [32]. However, chronic and non-resolute inflammations exist, which impair satellite cell and skeletal muscle differentiation [33,34]. Perandini et al. reported that an imbalance in M1 and M2 macrophage expression in chronic inflammation impairs satellite cell and skeletal muscle differentiation [20]. Therefore, periodontitis-induced chronic inflammation has been suggested to affect satellite cell differentiation. 

However, no significant difference was observed in the CSA of myotubes. Tang et al. reported that myofiber CSA changes were observed at eight weeks in rats [35]. Therefore, the intervention duration may have been too short to evaluate muscle hypertrophy in this study. In addition, satellite cells may not affect muscle hypertrophy; however, they play an important role in muscle repair [36]. However, *P. gingivalis* is one of the most important pathogens of periodontal disease. Recent studies have shown that *P. gingivalis* is present in adolescents and young adults who are clinically healthy without periodontal lesions [14]. This indicates that an increase of *P. gingivalis* caused by bad oral health can affect the skeletal muscle not only in patients with periodontitis but also young people who are clinically healthy without periodontal lesions, such as young athletes. Therefore, keeping good oral health is important to skeletal muscle performance in all generations. In this study, we only analyzed the relationship between *P. gingivalis* and skeletal muscle. In future studies, analyzing the effect of other pathogens of periodontal disease (*Treponema denticola*, *Tannerella forsythensis*) on skeletal muscle or the relationship between oral microbial dysbiosis and skeletal muscle regeneration may help to clarify the effect of “periodontitis” on skeletal muscle. 

## 5. Conclusions

We evaluated the effects of the oral administration of *P. gingivalis* in vivo in rat skeletal muscle and suggested that *P. gingivalis* likely interferes with muscle repair following training. This result suggests that the disorganization of *P. gingivalis* bacteremia, one of the most common pathogens of periodontal disease, may help people achieve smoother training effects.

## Figures and Tables

**Figure 1 ijerph-19-13046-f001:**
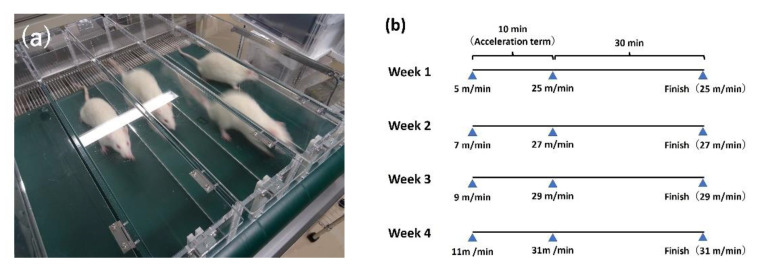
(**a**) Treadmill training; (**b**) protocol of treadmill training.

**Figure 2 ijerph-19-13046-f002:**
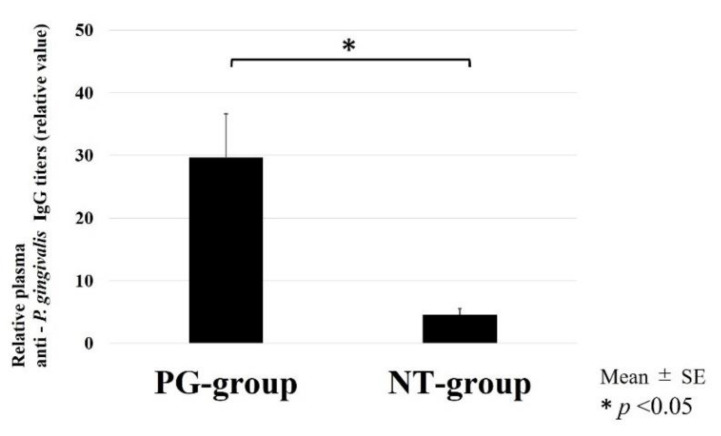
Serum IgG antibody titers in response to *P. gingivalis*.

**Figure 3 ijerph-19-13046-f003:**
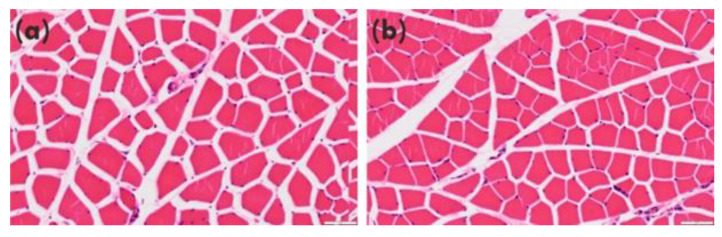
Histology of the myotubes: (**a**) PG group; (**b**) NT group; (**c**) average CSA of both groups. There were no significant differences between the groups.

**Figure 4 ijerph-19-13046-f004:**
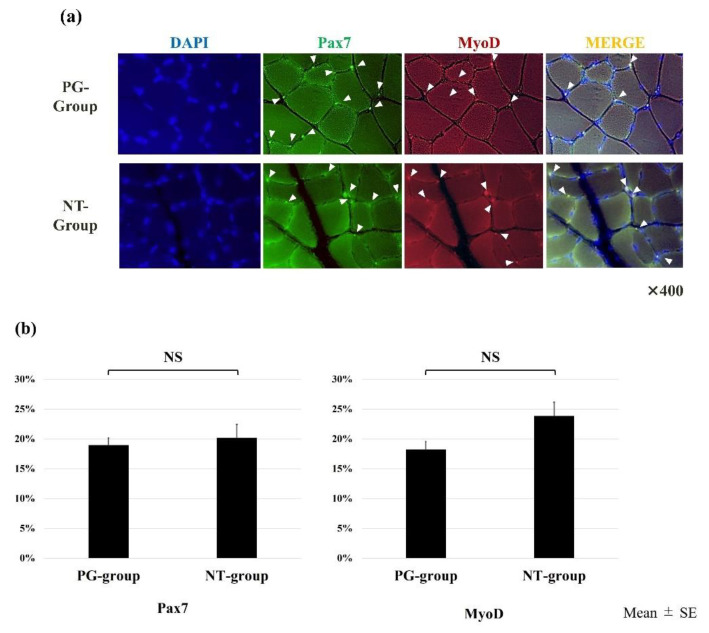
(**a**) Expression of Pax7 and MyoD; (**b**) the rates of Pax7- and MyoD-positive cells; (**c**) the results of Pax7/MyoD double staining. The Pax7−/MyoD+ cell was significantly larger in the PG group than in the NT group.

**Figure 5 ijerph-19-13046-f005:**
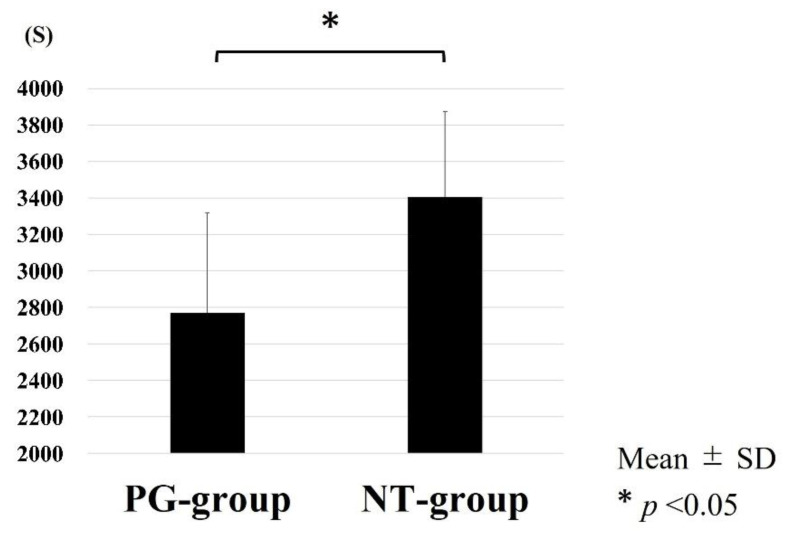
Time to running exhaustion. Running time of the PG group is significantly lower than that of the NT group.

**Table 1 ijerph-19-13046-t001:** Body and muscle wet weight (Mean ± SD). There was no significant difference among the two groups.

	Body–Before (g)	Body–After (g)	Gastrocnemius (g)
PG-group	178.26 (±20.77)	251.23 (±19.72)	1.628 (±0.140)
NT-group	180.69 (±23.15)	257.77 (±22.10)	1.668 (±0.111)
*p*-value	0.762	0.450	0.820

## Data Availability

The dataset can be accessed from the corresponding author upon reasonable request.

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
