# Peer review of "Continuous Oral Administration of Sonicated P. gingivalis Delays Rat Skeletal Muscle Healing Post-Treadmill Training"

_ijerph, 2022, doi:10.3390/ijerph192013046_

Round 1
Reviewer 1 Report
In this study, Hayashi et al evaluated the potential relationship between P. gingivalis administration and muscle regeneration after training. The authors performed several morphological and quantitative analysis and successfully concluded that infection with P. gingivalis inhibits muscle repair process after training. This would be a significant study because, as the authors described, little has been known whether/how P.gingivalis would inhibit muscle repair process using animal model. However, the manuscript and data could be still unmature and needed to be revised. Specific comments from this reviewer are as following.
Major comments
1. Whole results section would be better to be edited. It would be better to add ‘rationale’ and ‘conclusion’ to each paragraph of the results section. Current descriptions are just the explain about the data, and this would not be sufficient enough. Please refer the example below.
Example; 3.1. Serum IgG antibody titers in response to P.gingivalis
To examine ……., we analyzed…… (Rationale). The serum IgG antibody titers in response to P. gingivalis in the PG and NT groups 160 are shown in Figure 2. The PG group had significantly higher scores than did the NT 161 group (p = 0.0342). In the PG group, IgG antibody titers were approximately 30 times 162 greater than those in the control serum. These data suggest that… (Conclusion).
2. It would be better to discuss about the potential mechanisms how does P. gingivalis inhibit muscle repair process. It would be great to add some histological/quantitative data showing inflammatory cytokines as the authors described in Introduction (Page 2, Line 55).
3. About Figure 4a, it would be great to add high magnification images showing the representative Pax7+/MyoD-, Pax7+/MyoD+, and Pax7-/MyoD+ cells. Current images are not clear enough.
4. Do the authors have any specific reason why they used the treadmill training in this study?
Minor comments
1. It would be helpful for readers to understand the methods if the authors add some schemas or picture of treadmill equipment to Figure 1.
2. How hard is this treadmill training (Page 2, Line 59) generally? Is that normal exercises? Or the professional training for athletes? Could the author clarify that?
3. How did the authors analyze CSAs? What was ROI? It would be nice to describe more detail about that.
4. Do the rats treated with P. gingivalis have any periodontal diseases or any other inflammatory diseases?
5. How about other pathogens of periodontal diseases (Aa, Tf, Td etc.)? Could you discuss about that?
Author Response
Dear Reviewer 1
We thank you for your thoughtful suggestions and insights. The manuscript has benefited from these insightful suggestions. I look forward to working with you move this manuscript closer to publication in the International journal of Environmental Research and Public Health.
The manuscript has been rechecked and the necessary changes have been made in accordance with the reviewers’ suggestions. Edited part while referencing your suggestion highlighted in red color. The responses to all comments have been prepared and attached herewith below.
Major comments
- Whole results section would be better to be edited. It would be better to add ‘rationale’ and ‘conclusion’ to each paragraph of the results section. Current descriptions are just the explain about the data, and this would not be sufficient enough. Please refer the example below.
Example; 3.1. Serum IgG antibody titers in response to P.gingivalis
To examine ……., we analyzed…… (Rationale). The serum IgG antibody titers in response to P. gingivalis in the PG and NT groups 160 are shown in Figure 2. The PG group had significantly higher scores than did the NT 161 group (p = 0.0342). In the PG group, IgG antibody titers were approximately 30 times 162 greater than those in the control serum. These data suggest that… (Conclusion).
Response:
I have edited to result section like your suggestion. Edited part is highlighted in red.
- It would be better to discuss about the potential mechanisms how does P. gingivalis inhibit muscle repair process. It would be great to add some histological/quantitative data showing inflammatory cytokines as the authors described in Introduction (Page 2, Line 55).
Response:
I have added the “For example, Zhang et al. [16] reported that 41% of patients with periodontal disease (probing periodontal pocket depth ≥ 4 mm) had elevated serum C-reactive protein lev-el (≥ 5 mg/L), compared to 19% of non-periodontal disease patients [16]. “ in line 63-66.
- About Figure 4a, it would be great to add high magnification images showing the representative Pax7+/MyoD-, Pax7+/MyoD+, and Pax7-/MyoD+ cells. Current images are not clear enough.
Response:
I have changed the figure 4a as high magnification images.
- Do the authors have any specific reason why they used the treadmill training in this study?
Response:
We use treadmill to give rat same intensity. So, I have added “All rats were exercised using a treadmill and both groups performed exercises of the same intensity.” in line 251-252.
Minor comments
- It would be helpful for readers to understand the methods if the authors add some schemas or picture of treadmill equipment to Figure 1.
Response:
I have added the treadmill picture in Figure1.
- How hard is this treadmill training (Page 2, Line 59) generally? Is that normal exercises? Or the professional training for athletes? Could the author clarify that?
Response:
I have added “ The exercise intensity in this study was assumed to be hard training, such as training for athletes.” in line 278-259.
- How did the authors analyze CSAs? What was ROI? It would be nice to describe more detail about that.
Response:
I have added “We traced the shapes of muscle fibers and calculated the CSA (µm2) using the known distance and detected pixel size. All measured CSA were averaged for each group.” In line 139-141.
- Do the rats treated with P. gingivalis have any periodontal diseases or any other inflammatory diseases?
Response:
We did not observe the any symptom of periodontal disease or inflammatory disease. However, IgG titers in response to P.gingivalis significantly increased. This results means oral administration of sonicated P. gingivalis effect whole body.
So, I have added “These data suggest that oral administration of P. gingivalis rose serum IgG titers in response to P. gingivalis in the blood of the whole body.” in line 179-180.
- How about other pathogens of periodontal diseases (Aa, Tf, Td etc.)? Could you discuss about that?
Response:
We only analyze P. gingivalis inthis study. But analyzing the effect of other pathogens of other periodontal disease and microvial dysbiosis.
So, I have added “In this study, we only analyzed the relationship between P. gingivalis and skeletal muscle. In future studies, analyzing the effect of other pathogens of periodontal dis-ease (Treponema denticola, Tannerella forsythensis) to skeletal muscle, or the relationship between oral microbial dysbiosis and skeletal muscle regeneration, may help to clarify the effect of “periodontitis” on skeletal muscle.” in line 289-293.
Thank you for your consideration. I look forward to hearing from you.
Sincerely,
Kairi Hayashi

Reviewer 2 Report
Original article, well written with a quality methodology. The subject is innovative and can initiate further research with applications in the sports field, in particular, or even in the elderly population.
Logically, however, it poses a number of problems:
1/ Unless I am mistaken, I cannot find any previous in vivo studies on P. gingivalis and skeletal muscle regeneration. If this is the case, this fact should be highlighted in your introduction.
2/ You rightly target P. gingivalis in your study. However, P. gingivalis should be placed in a broader vision such as the Socransky complex and the periodontal microbiota with the passage from symbiosis to dysbiosis. P. gingivalis is "nothing" without the periodontal community and the host's responses. The risk is to suggest in a journal such as IJERPH vs. a periodontal journal that solving the P. gingivalis problem alone can advance the "cause".
3/ 2.2 Induction of experimental periodontal disease. I do not see how injecting P. gingivalis induces periodontal disease. This is a rather outdated concept. Yes, however, in the logic of inducing an inflammatory, immune reaction, etc. by the contribution of P. gingivalis.
4/ I have a doubt about the objective of the study which is not clearly specified in the introduction ("This study aimed to evaluate the effects of one of the most common pathogens of 224 periodontitis on muscle regeneration in vivo")
Please does your analysis really target the regeneration process or a degradation process?
5/ An important point. You reason in terms of periodontal disease. This is correct but the scope of your hypothesis goes beyond periodontal disease. Let me explain. Recent studies show that P. gingivalis and the main periodontal bacteria are significantly present in adolescence and in young adults who are clinically healthy without periodontal lesions. The problem of the interdental space, whose biofilm is never disorganized mechanically (brushing) or chemically, is put forward. We are in the concept of para-inflammation with a blood diffusion of Socransky's bacteria from the earliest age. As a reminder, the inter papillary gingiva is not keratinized and gingivitis (bleeding) has a high incidence.
In practice? Wouldn't the presence of these pathogenic bacteria impact muscle performance? I invite you to read the reference articles: DOI: 10.1371/journal.pone.0185804 and DOI: 10.3389/fmicb.2016.00840
6/ Conclusion: One should weigh up
We evaluated the effects of oral (??) administration of P. gingivalis in vivo (in Rat Skeletal Muscle ) and demonstrated (Too much) that contraction of periodontitis (No it's Pg) likely interferes with muscle repair following training. This result suggests that the prevention (not prevention. Perhaps disorganization) of P. gingivalis bacteremia, one of the most common pathogens of periodontal disease, may help people keep good health (It is not you goal???)
Prospective. Recommendations please for next researches.
References: 9/34 references are about 20 years old. Is this normal? This penalizes your topic
Author Response
Dear Reviewer 2
We thank you for your thoughtful suggestions and insights. The manuscript has benefited from these insightful suggestions. I look forward to working with you move this manuscript closer to publication in the International journal of Environmental Research and Public Health.
The manuscript has been rechecked and the necessary changes have been made in accordance with the reviewers’ suggestions. Edited part while referencing your suggestion highlighted in purple color. The responses to all comments have been prepared and attached herewith below.
Comments and Response
1/ Unless I am mistaken, I cannot find any previous in vivo studies on P. gingivalis and skeletal muscle regeneration. If this is the case, this fact should be highlighted in your introduction.
Response:
I have added “This is the first study to analyze the relationship between P. gingivalis and skeletal muscle in vivo.” in line 75-76.
2/ You rightly target P. gingivalis in your study. However, P. gingivalis should be placed in a broader vision such as the Socransky complex and the periodontal microbiota with the passage from symbiosis to dysbiosis. P. gingivalis is "nothing" without the periodontal community and the host's responses. The risk is to suggest in a journal such as IJERPH vs. a periodontal journal that solving the P. gingivalis problem alone can advance the "cause".
Response:
Just as you said, this study only clarified the relationship between skeletal muscle and “P. gingivalis” not “periodontitis”. I have edited all article in mind such as “Recent studies have suggested that P. gingivalis is a “keystone” of oral microbial dysbiosis, in which P. gingivalis acts as a critical agent by disrupting host immune homeostasis, and may lead to patient periodontitis. P. gingivalis is also present in adoles-cents and young adults without periodontal disease, and does not lead to periodontitis independently. However, some conditions such as bad oral health can accelerate gin-givitis or bone loss, resulting in an increase in oral P. gingivalis [13], and many bacterial toxins and components spread to the whole body through the blood stream.” in line 54-61.
3/ 2.2 Induction of experimental periodontal disease. I do not see how injecting P. gingivalis induces periodontal disease. This is a rather outdated concept. Yes, however, in the logic of inducing an inflammatory, immune reaction, etc. by the contribution of P. gingivalis.
Response:
I have added “The PG group rats were orally administered 200 µL of saline with sonicated P. gingi-valis using micropipette four times per week for six weeks.” in line 93-96.
4/ I have a doubt about the objective of the study which is not clearly specified in the introduction ("This study aimed to evaluate the effects of one of the most common pathogens of 224 periodontitis on muscle regeneration in vivo")
Please does your analysis really target the regeneration process or a degradation process?
Response:
I have changed “This study aimed to evaluate the effects of P. gingivalis, one of the most common pathogens of periodontitis, on muscle degradation in vivo.” In line 244-245.
5/ An important point. You reason in terms of periodontal disease. This is correct but the scope of your hypothesis goes beyond periodontal disease. Let me explain. Recent studies show that P. gingivalis and the main periodontal bacteria are significantly present in adolescence and in young adults who are clinically healthy without periodontal lesions. The problem of the interdental space, whose biofilm is never disorganized mechanically (brushing) or chemically, is put forward. We are in the concept of para-inflammation with a blood diffusion of Socransky's bacteria from the earliest age. As a reminder, the inter papillary gingiva is not keratinized and gingivitis (bleeding) has a high incidence.
In practice? Wouldn't the presence of these pathogenic bacteria impact muscle performance? I invite you to read the reference articles: DOI: 10.1371/journal.pone.0185804 and DOI: 10.3389/fmicb.2016.00840
Response:
Thank you for suggest the reference article. These articles helped me to understand periodontal disease. The mentioned above, I have edited manuscript as the relationship of skeletal muscle and P.gingivalis, not periodontitis.
6/ Conclusion: One should weigh up
We evaluated the effects of oral (??) administration of P. gingivalis in vivo (in Rat Skeletal Muscle ) and demonstrated (Too much) that contraction of periodontitis (No it's Pg) likely interferes with muscle repair following training. This result suggests that the prevention (not prevention. Perhaps disorganization) of P. gingivalis bacteremia, one of the most common pathogens of periodontal disease, may help people keep good health (It is not you goal???)
Response:
I have changed conclusion as “We evaluated the effects of oral administration of P. gingivalis in vivo in rat skel-etal muscle and suggested that P. gingivalis likely interferes with muscle repair fol-lowing training. This result suggests that the disorganization of P. gingivalis bactere-mia, one of the most common pathogens of periodontal disease, may help people achieve smoother training effects.” in line 296-300.
Prospective. Recommendations please for next researches.
Response:
We should do more study to clarify the relationship between skeletal muscle and “periodontitis.
So, I have added “n this study, we only analyzed the relationship between P. gingivalis and skeletal mus-cle. In future studies, analyzing the effect of other pathogens of periodontal disease (Treponema denticola, Tannerella forsythensis) to skeletal muscle, or the relationship be-tween oral microbial dysbiosis and skeletal muscle regeneration, may help to clarify the effect of “periodontitis” on skeletal muscle.” In line 290-294.
References: 9/34 references are about 20 years old. Is this normal? This penalizes your topic
Response:
Some references were changed to younger one. But some references were about 20years old, but very important one.
Thank you for your consideration. I look forward to hearing from you.
Sincerely,
Kairi Hayashi

Round 2
Reviewer 2 Report
Thank you for your modification
I'm sorry but we seem to have misunderstood each other. It is the title 3/2.2 that I have a problem with. In what way does the administration of P. gingivalis induce periodontal disease? No level of evidence for the term "periodontal disease" (please change the term i.e. inflammation, etc...
Line 55-56:Reference is missing
Line 56-57: Reference is missing
Author Response
Dear Reviewer 2 (Round 2)
Thank you for your thoughtful suggestions and insights. I misunderstood your indication. I’m sorry.
I have edited manuscript in accordance with your suggestions again. Edited part highlighted in blue. The responses to all comments have been prepared and attached herewith below.
Comment:
I'm sorry but we seem to have misunderstood each other. It is the title 3/2.2 that I have a problem with. In what way does the administration of P. gingivalis induce periodontal disease? No level of evidence for the term "periodontal disease" (please change the term i.e. inflammation, etc...
Response:
I have changed the title of subhead 2.2 as “Administration of P. gingivalis” in line 88.
And, I have added “It was considered that oral administration of P. gingivalis may cause it to invade the blood stream through the gingival crevice.” in line 250-251.
Comment:
Line 55-56: Reference is missing
Response:
I have added reference no.13.
Comment:
Line 56-57: Reference is missing
Response:
I have added reference no.14.
Thank you for your consideration. I look forward to hearing from you.
Sincerely,
